# Mixing Dirichlet Topic Models and Word Embeddings to Make lda2vec

## Abstract

Distributed dense word vectors have been shown to be effective at capturing token-level semantic and syntactic regularities in language, while topic models can form interpretable representations over documents. In this work, we describe *lda2vec*, a model that learns dense word vectors jointly with Dirichlet-distributed latent document-level mixtures of topic vectors. In contrast to continuous dense document representations, this formulation produces sparse, interpretable document mixtures through a non-negative simplex constraint. Our method is simple to incorporate into existing automatic differentiation frameworks and allows for unsupervised document representations geared for use by scientists while simultaneously learning word vectors and the linear relationships between them.

## 1 Introduction

Topic models are popular for their ability to organize document collections into a smaller set of prominent themes. In contrast to dense distributed representations, these document and topic representations are generally accessible to humans and more easily lend themselves to being interpreted. This interpretability provides additional options to highlight the patterns and structures within our systems of documents. For example, using Latent Dirichlet Allocation (LDA) topic models can reveal cluster of words within documents (Blei et al., 2003), highlight temporal trends (Charlin et al., 2015), and infer networks of complementary products (McAuley et al., 2015). See Blei et al. (2010) for an overview of topic modelling in domains as diverse as computer vision, genetic markers, survey data, and social network data.

Dense vector approaches to building document representations also exist: Le and Mikolov (2014) propose paragraph vectors that are predictive of bags of words within paragraphs, Kiros et al. (2015) build vectors that reconstruct the sentence sequences before and after a given sentence, and Ghosh et al. (2016) construct contextual LSTMs that predict proceeding sentence features. Probabilistic topic models tend to form documents as a sparse mixed-membership of topics while neural network models tend to model documents as dense vectors. By virtue of both their sparsity and low-dimensionality, representations from the former are simpler to inspect and more immediately yield high level intuitions about the underlying system (although not without hazards, see Chang et al. (2009)). This paper explores hybrid approaches mixing sparse document representations with dense word and topic vectors.

Unfortunately, crafting a new probabilistic topic model requires deriving a new approximation, a procedure which takes substantial expertise and must be customized to every model. As a result, prototypes are time-consuming to develop and changes to model architectures must be carefully considered. However, with modern automatic differentiation frameworks the practitioner can focus development time on the model design rather than the model approximations. This expedites the process of evaluating which model features are relevant. This work takes advantage of the Chainer (Tokui et al., 2015) framework to quickly develop models while also enabling us to utilize GPUs to dramatically improve computational speed.

Finally, traditional topic models over text do not take advantage of recent advances in distributed word representations which can capture semantically meaningful regularities between tokens. The

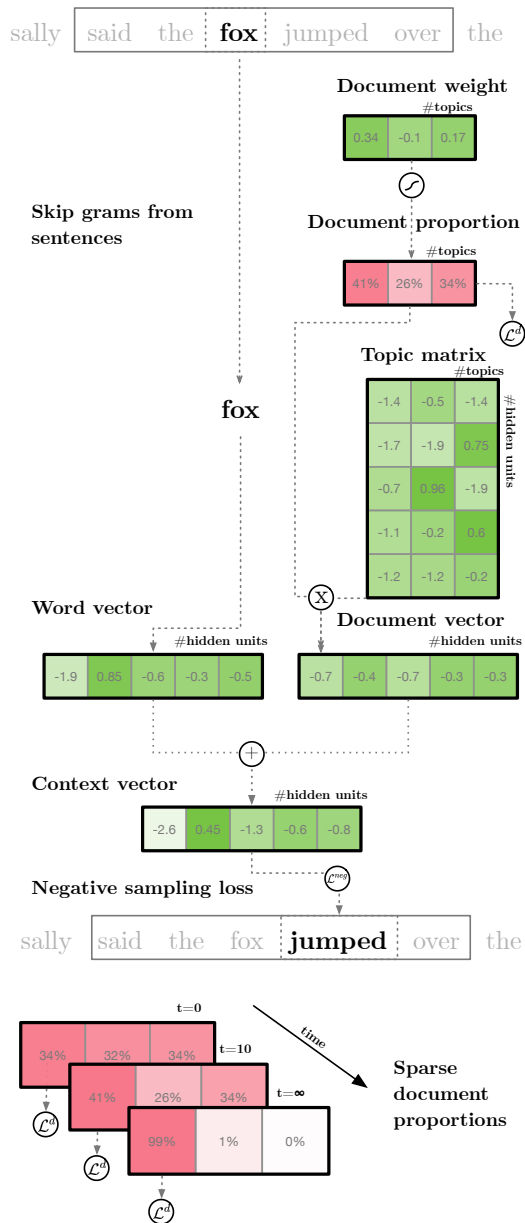

Figure 1: *lda2vec* builds representations over both words and documents by mixing word2vec's skip-gram architecture with Dirichlet-optimized sparse topic mixtures. The various components and transformations present in the diagram are described in the text.

examination of word co-occurrences has proven to be a fruitful research paradigm. For example, Mikolov et al. (2013) utilize Skipgram Negative-Sampling (SGNS) to train word embeddings using word-context pairs formed from windows moving across a text corpus. These vector representations ultimately encode remarkable linearities such as $king - man + woman = queen$. In fact, Levy and Goldberg (2014c) demonstrate that this is implicitly factorizing a variant of the Pointwise Mutual Information (PMI) matrix that emphasizes predicting frequent co-occurrences over rare ones. Closely related to the PMI matrix, Pennington et al. (2014) factorize a large global word count co-occurrence matrix to yield more efficient and slightly more performant computed embeddings than SGNS. Once created, these representations are then useful for information retrieval (Manning et al., 2009) and parsing tasks (Levy and Goldberg, 2014a). In this work, we will take advantage of word-level representations to build document-level abstractions.

This paper extends distributed word representations by including interpretable document representations and demonstrate that model inference can be performed and extended within the framework of automatic differentiation.

## 2 Model

This section describes the model for *lda2vec*. We are interested in modifying the Skipgram Negative-Sampling (SGNS) objective in (Mikolov et al., 2013) to utilize document-wide feature vectors while simultaneously learning continuous document weights loading onto topic vectors. The network architecture is shown in Figure 1.

The total loss term $\mathcal{L}$ in (1) is the sum of the Skipgram Negative Sampling Loss (SGNS) $\mathcal{L}_{ij}^{neg}$ with the addition of a Dirichlet-likelihood term over document weights, $\mathcal{L}^d$ that will be discussed later. The loss is conducted using a context vector, $\vec{c_j}$, pivot word vector $\vec{w_j}$, target word vector $\vec{w_i}$, and negatively-sampled word vector $\vec{w_l}$.

$$\mathcal{L} = \mathcal{L}^d + \Sigma_{ij}\mathcal{L}_{ij}^{neg} \quad (1)$$
$$\mathcal{L}_{ij}^{neg} = \log \sigma(\vec{c_j} \cdot \vec{w_i}) + \Sigma_{l=0}^n \log \sigma(-\vec{c_j} \cdot \vec{w_l}) \quad (2)$$

### 2.1 Word Representation

As in Mikolov et al. (2013), pairs of pivot and target words $(j, i)$ are extracted when they co-

occur in a moving window scanning across the corpus. In our experiments, the window contains five tokens before and after the pivot token. For every pivot-target pair of words the pivot word is used to predict the nearby target word. Each word is represented with a fixed-length dense distributed-representation vector, but unlike Mikolov et al. (2013) the same word vectors are used in both the pivot and target representations. The SGNS loss shown in (2) attempts to discriminate context-word pairs that appear in the corpus from those randomly sampled from a 'negative' pool of words. This loss is minimized when the observed words are completely separated from the marginal distribution. The distribution from which tokens are drawn is $u^\beta$, where $u$ denotes the overall word frequency normalized by the total corpus size. Unless stated otherwise, the negative sampling power $beta$ is set to $3/4$ and the number of negative samples is fixed to $n = 15$ as in Mikolov et al. (2013). Note that a distribution of $u^{0.0}$ would draw negative tokens from the vocabulary with no notion of popularity while a distribution proportional with $u^{1.0}$ draws from the empirical unigram distribution. Compared to the unigram distribution, the choice of $u^{3/4}$ slightly emphasizes choosing infrequent words for negative samples. In contrast to optimizing the softmax cross entropy, which requires modelling the overall popularity of each token, negative sampling focuses on learning word vectors conditional on a context by drawing negative samples from each token's marginal popularity in the corpus.

## 2.2 Document Representations

*lda2vec* embeds both words and document vectors into the same space and trains both representations simultaneously. By adding the pivot and document vectors together, both spaces are effectively joined. Mikolov et al. (2013) provide the intuition that word vectors can be summed together to form a semantically meaningful combination of both words. For example, the vector representation for $Germany + airline$ is similar to the vector for $Lufthansa$. We would like to exploit the additive property of word vectors to construct a meaningful sum of word and document vectors. For example, if as *lda2vec* is scanning a document the $j$th word is $Germany$, then neighboring words are predicted to be similar such as $France$, $Spain$, and $Austria$. But if the document is specifically

about airlines, then we would like to construct a document vector similar to the word vector for $airline$. Then instead of predicting tokens similar to $Germany$ alone, predictions similar to both the document and the pivot word can be made such as: $Lufthansa$, $Condor$ $Flugdienst$, and $Aero$ $Lloyd$. Motivated by the meaningful sums of words vectors, in *lda2vec* the context vector is explicitly designed to be the sum of a document vector and a word vector as in (3):

$$\vec{c_j} = \vec{w_j} + \vec{d_j} \qquad (3)$$

This models document-wide relationships by preserving $\vec{d_j}$ for all word-context pairs in a document, while still leveraging local inter-word relationships stemming from the interaction between the pivot word vector $\vec{w_j}$ and target word $\vec{w_i}$. The document and word vectors are summed together to form a context vector that intuitively captures long- and short-term themes, respectively. In order to prevent co-adaptation, we also perform dropout on both the unnormalized document vector $\vec{d_j}$ and the pivot word vector $\vec{w_j}$ (Hinton et al., 2012).

### 2.2.1 Document Mixtures

If we only included structure up to this point, the model would produce a dense vector for every document. However, *lda2vec* strives to form interpretable representations and to do so an additional constraint is imposed such that the document representations are similar to those in traditional LDA models. We aim to generate a document vector from a mixture of topic vectors and to do so, we begin by constraining the document vector $\vec{d_j}$ to project onto a set of latent topic vectors $\vec{t_0}, \vec{t_1}, ..., \vec{t_k}$:

$$\vec{d_j} = p_{j0} \cdot \vec{t_0} + p_{j1} \cdot \vec{t_1} + ... + p_{jk} \cdot \vec{t_k} + ... + p_{jn} \cdot \vec{t_n} \quad (4)$$

Each weight $0 \le p_{jk} \le 1$ is a fraction that denotes the membership of document $j$ in the topic $k$. For example, the Twenty Newsgroups model described later has 11313 documents and $n = 20$ topics so $j = 0...11312$, $k = 0...19$. When the word vector dimension is set to 300, it is assumed that the document vectors $\vec{d_j}$, word vectors $\vec{w_i}$ and topic vectors $\vec{t_k}$ all have dimensionality 300. Note that the topics $\vec{t_k}$ are shared and are a common component to all documents but whose strengths are modulated by document weights $p_{jk}$ that are

unique to each document. To aid interpretability, the document memberships are designed to be non-negative, and to sum to unity. To achieve this constraint, a softmax transform maps latent vectors initialized in $\mathbb{R}^{300}$ onto the simplex defined by $p_{jk}$. The softmax transform naturally enforces the constraint that $\Sigma_k p_{jk} = 1$ and allows us interpret memberships as percentages rather than unbounded weights.

Formulating the mixture in (4) as a sum ensures that topic vectors $\vec{t_k}$, document vectors $\vec{d_j}$ and word vectors $\vec{w_i}$, operate in the same space. As a result, what words $\vec{w_i}$ are most similar to any given topic vector $\vec{t_k}$ can be directly calculated. While each topic is not literally a token present in the corpus, it's similarity to other tokens is meaningful and can be measured. Furthermore, by examining the list of most similar words one can attempt to interpret what the topic represents. For example, by calculating the most similar token to any topic vector (e.g. $argmax_i(\vec{t_0} \cdot \vec{w_i})$) one may discover that the first topic vector $\vec{t_0}$ is similar to the tokens *pitching*, *catcher*, and *Braves* while the second topic vector $\vec{t_1}$ may be similar to *Jesus*, *God*, and *faith*. This provides us the option to interpret the first topic as *baseball* topic, and as a result the first component in every document proportion $p_{j0}$ indicates how much document $j$ is in the *baseball* topic. Similarly, the second topic may be interpreted as *Christianity* and the second component of any document proportion $p_{j1}$ indicates the membership of that document in the *Christianity* topic.

### 2.2.2 Sparse Memberships

Finally, the document weights $p_{ij}$ are sparsified by optimizing the document weights with respect to a Dirichlet likelihood with a low concentration parameter $\alpha$:

$$\mathcal{L}^d = \lambda \Sigma_{jk} (\alpha - 1) \log p_{jk} \qquad (5)$$

The overall objective in (5) measures the likelihood of document $j$ in topic $k$ summed over all available documents. The strength of this term is modulated by the tuning parameter $\lambda$. This simple likelihood encourages the document proportions coupling in each topic to be sparse when $\alpha < 1$ and homogeneous when $\alpha > 1$. To drive interpretability, we are interested in finding sparse memberships and so set $\alpha = n^{-1}$ where $n$ is the number of topics. We also find that setting

the overall strength of the Dirichlet optimization to $\lambda = 200$ works well. Document proportions are initialized to be relatively homogeneous, but as time progresses, the $\mathcal{L}^d$ encourages document proportions vectors to become more concentrated (e.g. sparser) over time. In experiments without this sparsity-inducing term (or equivalently when $\alpha = 1$) the document weights $p_{ij}$ tend to have probability mass spread out among all elements. Without any sparsity inducing terms the existence of so many non-zero weights makes interpreting the document vectors difficult. Furthermore, we find that the topic basis are also strongly affected, and the topics become incoherent.

### 2.3 Preprocessing and Training

The objective in (1) is trained in individual mini-batches at a time while using the Adam optimizer (Kingma and Ba, 2014) for two hundred epochs across the dataset. The Dirichlet likelihood term $\mathcal{L}^d$ is typically computed over all documents, so in modifying the objective to minibatches we adjust the loss of the term to be proportional to the minibatch size divided by the size of the total corpus. Our software is open source, available online, documented and unit tested[1]. Finally, the top ten most likely words in a given topic are submitted to the online *Palmetto*[2] topic quality measuring tool and the coherence measure $C_v$ is recorded. After evaluating multiple alternatives, $C_v$ is the recommended coherence metric in Röder et al. (2015). This measure averages the Normalized Pointwise Mutual Information (NPMI) for every pair of words within a sliding window of size 110 on an external corpus and returns mean of the NPMI for the submitted set of words. Token-to-word similarity is evaluated using the `3COSMUL` measure (Levy and Goldberg, 2014b).

## 3 Experiments

### 3.1 Twenty Newsgroups

This section details experiments in discovering the salient topics in the Twenty Newsgroups dataset, a popular corpus for machine learning on text. Each document in the corpus was posted to one of twenty possible newsgroups. While the text of each post is available to *lda2vec*, each of the

---

[1] The code for *lda2vec* is available online at `https://github.com/cemoody/lda2vec`

[2] The online evaluation tool can be accessed at `http://palmetto.aksw.org/palmetto-webapp/`

| # of topics | $\beta$ | Topic Coherences |
|---:|---:|---:|
| 20 | 0.75 | **0.567** |
| 30 | 0.75 | 0.555 |
| 40 | 0.75 | 0.553 |
| 50 | 0.75 | 0.547 |
| 20 | 1.00 | 0.563 |
| 30 | 1.00 | 0.564 |
| 40 | 1.00 | 0.552 |
| 50 | 1.00 | 0.558 |

Figure 2: Average topic coherences found by *lda2vec* in the Twenty Newsgroups dataset are given. The topic coherence has been demonstrated to correlate with human evaluations of topic models (Röder et al., 2015). The number of topics chosen is given, as well as the negative sampling exponent parameter $\beta$. Compared to $\beta = 1.00$, $\beta = 0.75$ draws more rare words as negative samples. The best topic coherences are found in models $n = 20$ topics and a $\beta = 0.75$.

newsgroup partitions is not revealed to the algorithm but is nevertheless useful for post-hoc qualitative evaluations of the discovered topics. The corpus is preprocessed using the data loader available in Scikit-learn (Pedregosa et al., 2012) and tokens are identified using the SpaCy parser (Honnibal and Johnson, 2015). Words are lemmatized to group multiple inflections into single tokens. Tokens that occur fewer than ten times in the corpus are removed, as are tokens that appear to be URLs, numbers or contain special symbols within their orthographic forms. After preprocessing, the dataset contains 1.8 million observations of 8,946 unique tokens in 11,313 documents. Word vectors are initialized to the pretrained values found in Mikolov et al. (2013) but otherwise updates are allowed to these vectors at training time.

A range of *lda2vec* parameters are evaluated by varying the number of topics $n \in 20, 30, 40, 50$ and the negative sampling exponent $\beta \in 0.75, 1.0$. The best topic coherences were achieved with $n = 20$ topics and with negative sampling power $\beta = 0.75$ as summarized in Figure 2. We briefly experimented with variations on dropout ratios but we did not observe any substantial differences.

Figure 3 lists four example topics discovered in the Twenty Newsgroups dataset. Each topic is associated with a topic vector that lives in the same space as the trained word vectors and listed are the most similar words to each topic vector. The first topic shown has high similarity to the tokens *astronomical*, *Astronomy*, *satellite*, *planetary*, and *telescope* and is thus likely a 'Space'-related topic similar to the 'sci.space' newsgroup. The second example topic is similar to words semantically related to 'Encryption', such as *Clipper* and *encrypt*, and is likely related to the 'sci.crypt' newsgroup. The third and four example topics are 'X Windows' and 'Middle East' which likely belong to the 'comp.windows.x' and 'talk.politics.mideast' newsgroups.

### 3.2 Hacker News Comments corpus

This section evaluates *lda2vec* on a very large corpus of Hacker News [3] comments. Hacker News is social content-voting website and community whose focus is largely on technology and entrepreneurship. In this corpus, a single document is composed of all of the words in all comments posted to a single article. Only stories with more than 10 comments are included, and only comments from users with more than 10 comments are included. We ignore other metadata such as votes, timestamps, and author identities. The raw dataset [4] is available for download online. The corpus is nearly fifty times the size of the Twenty Newsgroups corpus which is sufficient for learning a specialized vocabulary. To take advantage of this rich corpus, we use the SpaCy to tokenize whole noun phrases and entities at once (Honnibal and Johnson, 2015). The specific tokenization procedure[5] is also available online, as are the preprocessed datasets [6] results. This allows us to capture phrases such as *community policing measure* and prominent figures such as *Steve Jobs* as single tokens. However, this tokenization procedure generates a vocabulary substantially different from the one available in the Palmetto topic coherence tool and so we do not report topic coherences on this corpus. After preprocessing, the corpus contains 75 million tokens in 66 thousand documents with 110 thousand unique tokens. Unlike the Twenty Newsgroups analysis, word vectors are initialized randomly instead of using a library of pretrained vectors.

---

[3] See https://news.ycombinator.com/
[4] The raw dataset is freely available at https://zenodo.org/record/45901
[5] The tokenization procedure is available online at https://github.com/cemoody/lda2vec/blob/master/lda2vec/preprocess.py
[6] A tokenized dataset is freely available at https://zenodo.org/record/49899

| Topic Label | "Space" | "Encryption" | "X Windows" | "Middle East" |
|---|---|---|---|---|
| Top tokens | astronomical | encryption | mydisplay | Armenian |
| | Astronomy | wiretap | xlib | Lebanese |
| | satellite | encrypt | window | Muslim |
| | planetary | escrow | cursor | Turk |
| | telescope | Clipper | pixmap | sy |
| Topic Coherence | 0.712 | 0.675 | 0.472 | 0.615 |

Figure 3: Topics discovered by *lda2vec* in the Twenty Newsgroups dataset. The inferred topic label is shown in the first row. The tokens with highest similarity to the topic are shown immediately below. Note that the twenty newsgroups corpus contains corresponding newsgroups such as *sci.space*, *sci.crypt*, *comp.windows.x* and *talk.politics.mideast*.

| "Housing Issues" | "Internet Portals" | "Bitcoin" | "Compensation" | "Gadget Hardware" |
|---|---|---|---|---|
| more housing | DDG. | btc | current salary | the Surface Pro |
| basic income | Bing | bitcoins | more equity | HDMI |
| new housing | Google+ | Mt. Gox | vesting | glossy screens |
| house prices | DDG | MtGox | equity | Mac Pro |
| short-term rentals | iGoogle | Gox | vesting schedule | Thunderbolt |

Figure 4: Topics discovered by *lda2vec* in the Hacker News comments dataset. The inferred topic label is shown in the first row. We form tokens from noun phrases to capture the unique vocabulary of this specialized corpus.

| Artificial sweeteners | Black holes | Comic Sans | Functional Programming | San Francisco |
|---|---|---|---|---|
| glucose | particles | typeface | FP | New York |
| fructose | consciousness | Arial | Haskell | Palo Alto |
| HFCS | galaxies | Helvetica | OOP | NYC |
| sugars | quantum mechanics | Times New Roman | functional languages | New York City |
| sugar | universe | font | monads | SF |
| Soylent | dark matter | new logo | Lisp | Mountain View |
| paleo diet | Big Bang | Anonymous Pro | Clojure | Seattle |
| diet | planets | Baskerville | category theory | Los Angeles |
| carbohydrates | entanglement | serif font | OO | Boston |

Figure 5: Given an example token in the top row, the most similar words available in the Hacker News comments corpus are reported.

We train an *lda2vec* model using 40 topics and 256 hidden units and report the learned topics that demonstrate the themes present in the corpus. Furthermore, we demonstrate that word vectors and semantic relationships specific to this corpus are learned.

In Figure 4 five example topics discovered by *lda2vec* in the Hacker News corpus are listed. These topics demonstrate that the major themes of the corpus are reproduced and represented in learned topic vectors in a similar fashion as in LDA (Blei et al., 2003). The first, which we hand-label *Housing Issues* has prominent tokens relating to housing policy issues such as housing supply (e.g. *more housing*), and costs (e.g. *basic income* and *house prices*). Another topic lists major internet portals, such as the privacy-conscious search engine 'Duck Duck Go' (in the corpus abbreviated as *DDG*), as well as other major search engines (e.g. *Bing*), and home pages (e.g. *Google+*, and *iGoogle*). A third topic is that of the popular online curency and payment system *Bitcoin*, the abbreviated form of the currency *btc*, and the now-defunct Bitcoin trading platform *Mt. Gox*. A fourth topic considers salaries and compensation with tokens such as *current salary*, *more equity* and *vesting*, the process by which employees secure stock from their employers. A fifth example topic is that of technological hardware like *HDMI* and *glossy screens* and includes devices such as *the Surface Pro* and *Mac Pro*.

Figure 5 demonstrates that token similarities are learned in a similar fashion as in SGNS (Mikolov et al., 2013) but specialized to the Hacker News corpus. Tokens similar to the token *Artificial sweeteners* include other sugar-related tokens like *fructose* and food-related tokens such as *paleo diet*. Tokens similar to *Black holes* include physics-related concepts such as *galaxies* and *dark matter*. The Hacker News corpus devotes a substantial quantity of text to fonts and design, and the words most similar to *Comic Sans* are other popular fonts (e.g. *Times New Roman* and *Helvetica*) as well as font-related concepts such as *typeface* and *serif font*. Tokens similar to *Functional Programming* demonstrate similarity to other computer science-related tokens while tokens similar to *San Francisco* include other large American cities as well smaller cities located in the San Francisco Bay Area.

Figure 6 demonstrates that in addition to learn-

| Query | Result |
|---|---|
| California + technology | Silicon Valley |
| digital + currency | Bitcoin |
| Javascript - browser + server | Node.js |
| Mark Zuckerberg - Facebook + Amazon | Jeff Bezos |
| NLP - text + image | computer vision |
| Snowden - United States + Sweden | Assange |
| Surface Pro - Microsoft + Amazon | Kindle |

Figure 6: Example linear relationships discovered by *lda2vec* in the Hacker News comments dataset. The first column indicates the example input query, and the second column indicates the token most similar to the input.

ing topics over documents and similarities to word tokens, linear regularities between tokens are also learned. The 'Query' column lists a selection of tokens that when combined yield a token vector closest to the token shown in the 'Result' column. The subtractions and additions of vectors are evaluated literally, but instead take advantage of the 3COSMUL objective (Levy and Goldberg, 2014b). The results show that relationships between tokens important to the Hacker News community exists between the token vectors. For example, the vector for *Silicon Valley* is similar to both *California* and *technology*, *Bitcoin* is indeed a *digital currency*, *Node.js* is a technology that enables running *Javascript* on *servers* instead of on client-side *browsers*, *Jeff Bezos* and *Mark Zuckerberg* are CEOs of *Amazon* and *Facebook* respectively, *NLP* and *computer vision* are fields of machine learning research primarily dealing with *text* and *images* respectively, Edward *Snowden* and Julian *Assange* are both whistleblowers who were primarily located in the *United States* and *Sweden* and finally the *Kindle* and the *Surface Pro* are both tablets manufactured by *Amazon* and *Microsoft* respectively. In the above examples semantic relationships between tokens encode for attributes and features including: location, currencies, server v.s. client, leadership figures, machine learning fields, political figures, nationalities, companies and hardware.

### 3.3 Conclusion

This work demonstrates a simple model, *lda2vec*, that extends SGNS (Mikolov et al., 2013) to build unsupervised document representations that yield coherent topics. Word, topic, and document vectors are jointly trained and embedded in a common representation space that preserves semantic regularities between the learned word vectors while still yielding sparse and interpretable document-to-topic proportions in the style of LDA (Blei et al., 2003). Topics formed in the Twenty Newsgroups corpus yield high mean topic coherences which have been shown to correlate with human evaluations of topics (Röder et al., 2015). When applied to a Hacker News comments corpus, *lda2vec* discovers the salient topics within this community and learns linear relationships between words that allow it solve word analogies in the specialized vocabulary of this corpus. Finally, we note that our method is simple to implement in automatic differentiation frameworks and can lead to more readily interpretable unsupervised representations.

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
