# Peer review of "Mixing Dirichlet Topic Models and Word Embeddings to Make lda2vec"

_CoNLL 2016 — decision unknown_

[Official Review · Reviewer 1 · rating 2 · confidence 4]
soundness 2 · originality 3 · clarity 1 · impact 3 · substance 4 · appropriateness 5 · meaningful comparison 5 · replicability 3 · presentation format Poster

A combination of word2vec and LDA could be potentially interesting. The main
problem with the current paper is that the technical details are
incomprehensible. Section 2 needs a complete rewrite so that a reader familiar
with word2vec and LDA could relatively easily get a high-level picture of how
the models are being combined. The current presentation doesn't achieve that.

More detailed comments:

The third paragraph of the introduction makes no sense to me. "requires
deriving a new approximation" - approximation of what? why is it time consuming
to develop prototypes? Why is it easier to evaluate features?

Why use the same word vectors for pivot and target (unlike in word2vec)? What's
the motivation for that decision?

what does it mean to separate words from a marginal distribution?

what's co-adaptation?

"If we only included structure up to this point" - what kind of structure?

"it's similarity" -> its

Footnote 1 breaks anonymity.

There doesn't appear to be any evaluation. The days when it was ok to just give
some example clusters are long gone in NLP. Figure 2 looks like it might be a
quantitative evaluation, but it's only described in the overly long caption.

The statement in the conclusion that the model solves word analogies is
overstating what was shown, which was just a few cherry-picked examples of king
+ queen etc. sort.

The Chang ref has the conference/journal name as "Advances in ..." You'd like
me to guess the venue?

[Official Review · Reviewer 2 · rating 2 · confidence 4]
soundness 4 · originality 3 · clarity 4 · impact 4 · substance 3 · appropriateness 5 · meaningful comparison 3 · replicability 4 · presentation format Oral Presentation

This paper proposes a neural-styled topic model, extending the objective of
word2vec to also learn document embeddings, which it then constrains through
sparsification, hence mimicking the output of a topic model.

I really liked the model that the authors proposed, and found the examples
presented by the authors to be highly promising. What was really missing from
the paper, however, was any empirical evaluation of the model -- evaluation
entirely falls back on tables of examples, without any indication of how
representative the examples are, or any attempt to directly compare with
standard or neural topic models. Without empirical evaluation, it is
impossible to get a sense of the true worth of the model, making it very hard
to accept the paper. Some ideas of how the authors could have achieved this:
(1) use the topic representation of each document in a supervised document
categorisation setup to compare against a topic model with the same topic
cardinality (i.e. as an indirect evaluation of the quality of the
representation); or (2) through direct evaluation over a dataset with document
similarity annotations (based on pairwise comparison over topic vectors).

It's fantastic that you are releasing code, but you have compromised anonymity
in publishing the github link in the submitted version of the paper (strictly
speaking, this is sufficient for the paper to be rejected outright, but I
leave that up to the PCs)

Other issues:

- how did you select the examples in Figures 3-6? presenting a subset of the
  actual topics etc. potentially reeks of cherry picking.

- in Section 2.2.1 you discuss the possibility of calculating word
  representations for topics based on pairwise comparison with each word in
  the vocabulary, but this is going to be an extremely expensive process for a
  reasonable vocab size and number of topics; is this really feasible?

- you say that you identify "tokens" using SpaCy in Section 3.1 -- how? You
  extract noun chunks (but not any other chunk type), similarly to the Section
  3.2, or something else? Given that you go on to say that you use word2vec
  pre-trained embeddings (which include only small numbers of multiword
  terms), it wasn't clear what you were doing here.

- how does your model deal with OOV terms? Yes, in the experiments you report
  in the paper you appear to train the model over the entire document
  collection so it perhaps isn't an immediate problem, but there will be
  contexts where you want to apply the trained model to novel documents, in
  which case the updating of the word2vec token embeddings is going to mean
  that any non-updated (OOV, relative to the training collection) word2vec
  embeddings are not going to be directly comparable to the tuned embeddings.

- the finding that 20 topics worked best over the 20 Newsgroups corpus wasn't
  surprising given its composition. Possibly another (very simple) form of
  evaluation here could have been based on some information-theoretic
  comparison relative to the true document labels, where again you would have
  been able to perform a direct comparison with LDA etc.

- a couple of other neural topic models that you meed to compare yourself with
  are:

Cao, Ziqiang, Sujian Li, Yang Liu, Wenjie Li, and Heng Ji. "A Novel Neural
Topic Model and Its Supervised Extension." In AAAI, pp. 2210-2216. 2015.

Nguyen, Dat Quoc, Richard Billingsley, Lan Du, and Mark Johnson. "Improving
Topic Models with Latent Feature Word Representations." Transactions of the
Association for Computational Linguistics 3 (2015): 299-313.

Shamanta, Debakar, Sheikh Motahar Naim, Parang Saraf, Naren Ramakrishnan, and
M. Shahriar Hossain. "Concurrent Inference of Topic Models and Distributed
Vector Representations." In Machine Learning and Knowledge Discovery in
Databases, pp. 441-457. Springer International Publishing, 2015.

Low-level things:

line 315: "it's similarity" -> "its similarity"

line 361: what does it mean for the "topic basis" to be affected (and the
"are" is awkward here)

- in the caption of Figure 5, the examples should perhaps be "terms" rather
  than "words"

- the reference formatting is all over the place, e.g. "Advances in ...",
  "Advances in Neural ...", Roder et al. is missing the conference name, etc.